# Keep Moving to Retain the Healthy Self: The Influence of Physical Exercise in Health Anxiety among Chinese Menopausal Women

**DOI:** 10.3390/bs13020140

**Published:** 2023-02-07

**Authors:** Huilin Wang, Qingxing Zhang, Yapeng Lin, Yang Liu, Ziqing Xu, Jingyu Yang

**Affiliations:** 1School of Physical Education, Hunan University of Science and Technology, Xiangtan 411201, China; 2Faculty of Economics, Chulalongkorn University, Bangkok 10330, Thailand; 3Graduate School of Human Sciences, Assumption University, Bangkok 10240, Thailand; 4Journalism School of Fanchangjiang, Neijiang Normal University, Neijiang 641100, China; 5National Institute of Development Administration, International College, Bangkok 10240, Thailand; 6Department of Medical Bioinformatics, University of Göttingen, 37077 Göttingen, Germany

**Keywords:** menopausal women, physical exercise, health anxiety, interpersonal competence, emotional intelligence

## Abstract

Menopause is a period of high incidence of chronic diseases. Women experience various physical and psychological discomforts during menopause, and hormonal changes exacerbate mood swings in menopausal women and also cause them to begin to experience excessive worry and anxiety about their health problems. This study was a cross-sectional survey investigating the relationship between physical activity and women’s health anxiety. Using cluster sampling, a valid sample of 455 females aged 45–50 was collected from 78 communities in five municipal districts in Changsha, China, and AMOS v.23 was adopted to construct a structural equation model to verify the hypotheses. The results indicate that interpersonal competence and emotional intelligence are negatively associated with health anxiety. Furthermore, interpersonal competence and emotional intelligence mediate the relationship between physical exercise and health anxiety, which means that menopausal women with more physical exercise, higher interpersonal competence, and higher emotional intelligence reported lower health anxiety. Finally, to alleviate menopausal women’s health anxiety and reduce their risk of chronic diseases, the government, community, and family should create conditions and opportunities for women to participate in group physical activities.

## 1. Introduction

Menopause is a period of high incidence of various chronic diseases in women, and physical and psychological discomfort aggravate women’s painful experiences during this period. As women reach menopause, a variety of physical and psychological symptoms begin to appear, including palpitations, hot flashes, night sweats, frequent urination, sleep disturbance, fatigue, depression, anxiety, and irritability [1]. It is estimated that more than 50% of women aged 40–55 have emotional problems such as nervousness or irritability [2]. Numerous studies have confirmed an association between menopause and increased depression and anxiety, as women experience more significant mood swings during this period caused by endocrine and physiological changes [3]. However, in addition to physiological factors, the increased life stress and decreased physical health of menopausal women during this period are considered by some scholars to explain this problem [4]. Without intervention, women are at increased risk of developing chronic diseases in the long run [5].

Regular physical exercise is considered the most beneficial intervention for physical and mental health [6]. Physical exercise is advocated in the medical field as essential for treatment and rehabilitation in many chronic diseases [7]. Given the low cost and low risk of physical exercise, such interventions are considered useful options for treating anxiety disorders [8]. Improving national health has become one of China’s national strategies, and several policies have been developed, including the “National Fitness Program (2021–2025)” and “Healthy China 2030” [9]. The Chinese government fully supports people’s participation in physical exercise in terms of policy and finance. It is proposed that by 2025, the per capita sports area will reach 2.6 square meters, and the proportion of people who regularly participate in physical exercise will reach 38.5% [10]. Square dancing, Taijiquan, Badu Anjin, and other physical activities with Chinese characteristics and popular among middle-aged and older adults have received the government’s attention [11]. Competitions in these activities are held in various places and communities, driving increased physical activity in middle-aged and older adults [12].

However, there are times when governments develop good policies with little effect [13]. There are many reasons for this, such as the lack of clear policy objectives or insufficient implementation at the grassroots level. Therefore, promptly reporting specific cases and problems to the government and relevant departments is an effective way to improve policy implementation [14]. In this study, the researchers believe that the government has developed good policies to encourage people to increase physical exercise to improve their health. It has also increased financial investment in sports services and facilities for the public. However, questions remain as to the actual levels of participation of menopausal women in sports activities, whether they have encountered barriers to the involvement process, and whether the government can help solve any such barriers. If gaps found to exist between ideals and reality can be reported in time, then, on the one hand, the efforts made by the government in the “National Fitness Program” can be better rewarded, and on the other hand, the difficulties encountered by menopausal women will be taken seriously and helped.

Existing research on anxiety in menopausal women typically addresses anxiety in a broad sense—panic disorder, social phobia, or generalized anxiety (e.g., excessive and uncontrollable worrying, irritability) [15]. Academia lacks in-depth research on specific anxiety in menopausal women and effective intervention recommendations. The researchers in this study believe that the anxiety of menopausal women mainly focuses on health anxiety, and that they are unable to adapt to the changes in their physical and mental health and sometimes even suspect that some of the discomfort they experience is a sign that they are suffering from a significant disease. In addition, unlike previous studies, this study has found that menopausal women are not afraid of socializing. Indeed, they are very eager to socialize, and the exchange of health information and knowledge with women of the same age can help alleviate their fear of illness. Based on the above research gaps and problems, this study proposed the following research objectives: (1) to understand the health anxiety of Chinese menopausal women; (2) to explore the relationship between physical exercise, interpersonal competence, emotional intelligence, and health anxiety of menopausal women; and (3) to report existing problems and make recommendations to the government and community.

In this study focused on the health anxiety of menopausal women, considering the effect of physical health on mental health, physical exercise was proposed as an intervention measure to relieve women’s health anxiety. The specific paths analyzed were that group physical exercise (e.g., square dancing, Taijiquan, Baduanjin) is beneficial for increasing social opportunities for menopausal women, thereby improving women’s social skills, and that social activities are advantageous to the development of women’s positive emotions and the improvement of their emotional control, thereby relieving health anxiety caused by physical discomfort. The study findings suggest that interpersonal competence and emotional intelligence mediate the relationship between physical exercise and health anxiety, illustrating that menopausal women who are more physically active, have more robust interpersonal competence, and have higher emotional intelligence exhibit lower health anxiety. Therefore, understanding the impact of physical exercise on the health anxiety of menopausal women will not only help the development of relevant theories but also help the government, community, and family to pay attention to the physical and mental health of menopausal women and to call on more people to join in physical exercise.

## 2. Literature Review and Hypothesis Development

### 2.1. The Concept of Variables

#### 2.1.1. Physical Exercise

The summary of physical exercise literature can be mainly summarized from two perspectives. First, from a macro perspective, physical activities refer to mass physical monitoring, sports events, the sports industry, daily physical exercise, and related sports activities [16]. From a microscopic perspective, physical exercise refers to daily physical exercise [17,18]. Because the samples in this study are menopausal women, the physical exercise defined in this study mainly refers to physical exercise from a microscopic perspective—that is, physical exercise performed to improve physical and mental health.

#### 2.1.2. Health Anxiety

In psychology and epidemiology, health anxiety is defined as worry about one’s health based on perceived health threats and misunderstanding of physical symptoms [19]. The definition also includes persistent fear that one may develop a severe illness, even after one has been cleared by a formal medical examination [20]. The health anxiety referred to in this study refers to the anxiety about health in women during menopause. In psychology, it is manifested in restlessness, emotional tension, and irritability [21]. In physiology, it manifests itself as hot flashes, sweating, irregular menstruation, dizziness, insomnia, light sleep, dreaminess, etc. [22]. Studies have shown that 50% to 60% of menopausal women have obvious anxiety symptoms [23]. Although the disease will not threaten life safety, it will seriously reduce patients’ quality of life, affect their physical and mental health, and bring heavy burdens to patients and their families [24].

#### 2.1.3. Interpersonal Competence

The study of interpersonal competence originated from the study of social competence. Regarding the definition of interpersonal competence, previous researchers have mainly approached it from two perspectives. There is the behavioral skills perspective, which defines interpersonal competence as communication skills, leadership skills, and skills for practical cooperation with others [25]. There is also the perspective of social cognition and behavioral performance, which defines interpersonal competence as individual social cognition and behavioral performance [26]. This study focuses on interpersonal competence from the perspective of social cognition and social performance.

#### 2.1.4. Emotional Intelligence

Emotional intelligence, first proposed by Salovey and Mayer in 1990, refers to the ability of individuals to monitor their own and others’ feelings and emotions, distinguish them, and use this information to guide their thinking and actions [27]. Bar-on [28] pointed out that emotional intelligence is the sum of a series of emotional skills, personality traits, and interpersonal abilities that affect people’s ability to cope with environmental needs and pressures. Law et al. [29], coming from a Chinese background, defined emotional intelligence as the ability to evaluate self-emotions, evaluate others’ emotions, use emotions, and regulate emotions. This article will also refer to Law et al.’s definition of emotional intelligence.

### 2.2. Hypotheses

#### 2.2.1. Interpersonal Competence, Emotional Intelligence, and Health Anxiety

Existing studies have shown that interpersonal competence has a significant impact on individuals’ mental health [30], self-awareness development [31], and ability to relieve anxiety and depression [32]. The research on the effects of interpersonal competence on emotional intelligence mainly focuses on regulating emotions [33], managing emotions, and understanding others’ feelings [34]. Individuals with strong social skills tend to maintain and develop good interpersonal relationships, leading to a better understanding of their emotions. For menopausal women, it requires more emotional resources to relieve their anxiety [35]. Emotional intelligence is the ability to regulate emotions and solve problems using acquired information [36]. Having low emotional intelligence will lead to long-term emotional imbalances, thus affecting one’s ability to cope with adverse emotions [37].

Meanwhile, the influence of interpersonal competence on anxiety mainly focuses on social and psychological anxiety [38]. According to previous studies, health anxiety is a specific manifestation of mental health [39]. Furthermore, studies have shown that interpersonal competence can provide individuals with a sense of security and more emotional support, which is conducive to relieving individual loneliness, improving personal happiness, and improving mental health [40]. Therefore, the following hypotheses are proposed in this study.

**Hypothesis 1** **(H1).***Interpersonal competence is positively related to emotional intelligence*.

**Hypothesis 2** **(H2).***Interpersonal competence is negatively related to health anxiety*.

**Hypothesis 3** **(H3).***Emotional intelligence is negatively related to health anxiety*.

#### 2.2.2. The Mediating Effects

Menopausal women are in a relatively sensitive emotional stage, with increasing pressure from work and life, resulting in certain negative emotions [41], such as jealousy, dissatisfaction, hostility, etc., which can also lead to interpersonal tension [42]. However, many studies have shown that physical exercise can cultivate their sense of social participation and their positive personality characteristics, enhance their social adaptability [43], reduce their anxiety, and improve their mental state. Through physical exercise, people can better communicate with others, overcome loneliness, relieve pressure, expand social communication, coordinate interpersonal relations, improve social adaptability, and promote healthy physical and mental development [44]. Especially beneficial is participation in group activities, such as square dancing, Tai chi, football, basketball, volleyball, and other activities [45]. Central to these activities is the communication process between people [46], making people feel free and relaxed and able to quickly form positive emotions [47,48].

Current research on the mediating role of interpersonal competence and emotional intelligence focuses on life satisfaction [49], anxiety [50], loneliness [51], and psychological health [52]. Meanwhile, some studies have found a relationship between physical exercise and interpersonal competence and some mental health problems [53]. Physical exercise can promote interpersonal interaction and improve interpersonal relationships [54] while fostering positive emotions in individuals [55]. At the same time, there is evidence that positive interpersonal competence can effectively improve personal anxiety [56]. When the individual has higher emotional intelligence, the ability to deal with their own emotions is more vital [57]. Thus, this study proposed the following mediation hypothesis.

**Hypothesis 4** **(H4).***Interpersonal competence and emotional intelligence mediate the relationship between physical exercise and health anxiety*.

A summary of all hypotheses is shown in Figure 1.

## 3. Methodology

### 3.1. Participants and Procedure

This study used a cluster sampling method and targeted menopausal women aged 40–55. After the study was approved by the ethics committee of the institution where the first author works (No. ECBSHUST 2022/0013), the researchers surveyed female residents aged 40–55 in 78 communities in five municipal districts in Changsha, China, from July to August 2022. All respondents were informed of the purpose of the survey and signed an informed consent form before completing the questionnaire. After completing the questionnaire, the respondents could choose one item from a range of daily necessities (e.g., toothpaste, washing gloves, garbage bags) as a reward. Six hundred questionnaires were distributed in the survey, 455 valid questionnaires were returned, and the valid questionnaire return rate was 75.8%.

Table 1 lists the demographic characteristics of the 455 menopausal women who participated in the survey. Among the respondents, (1) one-half were between the ages of 46 and 50; (2) in terms of menopausal status, 56.7% were in the late reproductive stage; (3) in terms of marital status, most were married (widows and divorcees were not included); (4) in terms of educational level, more than one-half had completed high school or vocational school education; (5) in terms of employment status, 63.1% were employed; (6) in terms of income, about one-half (49.5%) indicated that their monthly income was around CNY 3000–6000 (USD 417–835).

### 3.2. Instruments

The questionnaire consisted of six parts. The first part asked respondents to report demographic information, including age, menopausal status, marital status, education level, employment status, and income. The second part collected data on respondents’ physical exercise in the most recent week, using the three items of the scale developed by Andersen et al. [58]; the sample items included “During the previous week, how often did you engage in light physical exercise such as walking, light cleaning, raking the lawn, or lightly strenuous exercise such as yoga, bowling, or similar activities?” The third part collected data on respondents’ interpersonal competence, measured using the four items of the scale developed by Buhrmester et al. [59]; the sample items included “I can find and suggest things to do with new people whom I find interesting and attractive.” The fourth part collected data on respondents’ emotional intelligence, measured using three items on a scale developed by Law et al. [29]; the sample items included “I can control my temper so that I can handle difficulties rationally”. Finally, the fifth part collected data on respondents’ health anxiety, measured using the four items of the scale developed by Abramowitz et al. [60]; the sample items included “I am fearful of having a serious illness”. All four of the above scales were measured using a five-point Likert scale, where responses ranged from 1 (i.e., strongly disagree, never) to 5 (i.e., strongly agree, always).

The researchers modified some items of the scales to suit the Chinese cultural background and context. As a result, a pilot test was required to ensure the reliability of the revised scales [61]. The pilot test received 82 valid questionnaires, and the results showed that the Cronbach coefficients were all higher than 0.8, which proved that the appropriate modification of the scales by the researchers was reasonable.

### 3.3. Data Analysis

This study used AMOS v23 to construct a structural equation model (SEM) of how menopausal women alleviated health anxiety through physical exercise, employing a maximum likelihood (ML) estimation method to estimate the parameters of the model. A two-step modeling approach was used to evaluate the measurement and structural models [62]. Firstly, the reliability and validity of the model were comprehensively evaluated. Then the fitting coefficient and path coefficient of the hypothetical model were measured, and the existence of the mediation effect was tested.

The researchers examined common method variation (CMV) and compared the degrees of freedom and chi-square values of the difference between Model 1 and Model 2, according to the method recommended by Mossholder et al. [63]. The results showed that the chi-square value of Model 1 was 2361.517, the degree of freedom was 77, and the *p*-value was less than 0.001; the chi-square value of Model 2 was 127.556, the degree of freedom was 71, and the *p*-value was less than 0.001. This meant that the fit of Model 1 was proportional to that of Model 2. The results indicated that the one-factor structure did not exist, so the CMV had little impact on this study and could be ignored.

## 4. Results

### 4.1. Assessment of the Measurement Model Reliability and Validity

This study calculated Cronbach’s alpha and composite reliability (CR) coefficients for latent variables to examine reliability and discriminant validity [64]. As shown in Table 2, the Cronbach’s alpha coefficients of the variables were in the range of 0.917–0.965, all CR values were higher than 0.9, and the average variance extracted (AVE) of all variables was in the range 0.769–0.874. Therefore, all variables had high reliability and convergent validity. Furthermore, as shown in Table 3, all correlation coefficients were less than the square root of AVE, indicating that all variables had good discriminant validity.

### 4.2. Hypothesis Testing Results

First, the error and residual terms of the structural equation model did not show negative values, indicating that the model did not violate the estimation. Second, the goodness of fit between the data and the structural equation model was high (χ^2^/df = 1.976, GFI = 0.959, AGFI = 0.941, NFI = 0.980, CFI = 0.990, TLI = 0.987, RMSEA = 0.046), much better than the suggested value. Third, according to the results of the Pearson correlation in Table 3, there were significant correlations between the independent variable, mediators, and dependent variable, which supported the validation of the hypotheses. Fourth, the structural path model in Figure 2 shows that the relationship between interpersonal competence and emotional intelligence was statistically significant (*β* = 0.658, *p* < 0.001), supporting H1; that the relationship between interpersonal competence and health anxiety was statistically significant (*β* = −0.212, *p* < 0.01), supporting H2; and that the relationship between emotional intelligence and health anxiety was statistically significant (*β* −0.582, *p* < 0.001), supporting H3.

The researchers hypothesized that physical exercise affects health anxiety through two mediators: interpersonal competence and emotional intelligence. This study used the bootstrap method to test for a mediating effect [65]. The standardized results of the 95% confidence interval for the 5000-bootstrap sample are shown in Table 4: the absolute values of all Z values are more significant than 1.96, and there is no zero value within the 95% confidence interval. In addition, interpersonal competence and emotional intelligence significantly affected the relationship between physical exercise and health anxiety (standardized indirect effect = −0.506, *p* < 0.001), supporting H4. The findings showed that menopausal women with more physical exercise, higher interpersonal competence, and higher emotional intelligence reported lower health anxiety.

## 5. Discussion

### 5.1. Theoretical Contribution

This study makes the following contributions to the theoretical analysis of female menopause. First, existing research focuses on anxiety [3], depression [4], and loneliness [66] in menopausal women, and there is little research on the specific nature of anxiety in menopausal women. This study is the first to focus on the issue of menopausal women’s health anxiety, which is more targeted than existing research and enriches relevant theoretical research. The researchers believe that research on the specific anxieties of menopausal women can pinpoint the root of the problem and thus find a solution. As with previous studies, the researchers believe that menopausal women worry about getting sick [67]. However, going one step further than previous studies, the researchers point out that the anxiety of menopausal women stems from their health problems, and that a series of physiological changes during menopause may make women fear and worry about diseases. A lack of knowledge about changes in the body exacerbates such concerns [68]. Additionally, many menopausal women are beginning to retire, and the end of their working life has caused them to return to ordinary family life, such as caring for a working husband or grandchildren. If post-retirement menopausal women stay at home for a long time and do not go out, their fears and worries about diseases caused by physiological changes are even more unspoken and unshared.

Second, this study is the first to explore the relationship between physical exercise and health anxiety in menopausal women. The results showed a significant positive relationship between physical exercise and interpersonal competence/emotional intelligence (see Figure 2), supporting the findings of Ubago-Jiménez et al. [69] and Li and Meng [70]. Physical exercise has the most significant impact on emotional intelligence, followed by interpersonal competence. Furthermore, interpersonal competence and emotional intelligence mediate the relationship between physical exercise and health anxiety. As shown in Figure 2, these variables explained 58% of the variance in health anxiety. This study provides a good path for studying the relationship between physical exercise and health anxiety—that is, from the perspective of the interpersonal function of group physical exercise—and profoundly explores the role of interpersonal competence.

### 5.2. Practical Implications

Considering the positive impact of physical exercise on interpersonal competence and emotional intelligence, as well as the indirect effect of physical exercise on relieving menopausal women’s health anxiety, the government, community, and family should provide both material and spiritual support for menopausal women to participate in physical exercise. For example, the government should improve the urban walking environment and improve the accessibility of parks and sports facilities [71]; the community should strengthen the provision of leisure services and improve group participation and the variety of daily physical activities; family members should encourage menopausal women to participate in physical activities, and should take the initiative to undertake part of the heavy housework to give them more time for leisure or exercise outside. Lack of opportunities is considered one of the reasons preventing people from increasing their physical activity and decreasing sedentary behavior [72]. Therefore, support from the government, community, and family will provide more opportunities for physical activity in menopausal women.

For a long time, people have misunderstood the emotional response of menopausal women, thinking that the emotional changes of menopausal women are like “bombs” or “floods,” and indifference and avoidance are the most common coping methods adopted by family members. In recent years, the word “menopause” has been seen as a derogatory term and used as an incorrect label to explain women’s emotional responses to this period [73]. Menopausal women are often portrayed as hysterical women, emotionally out-of-control mothers, and unreasonable wives [74]. Such misunderstandings and discrimination not only fail to relieve the pressures experienced by menopausal women but also are more likely to cause disharmony among family members and cause society to stereotype menopausal women as women who are not easy to get along with. This study not only points out how the stress faced by menopausal women manifests itself as physical discomfort, but also encourages menopausal women to participate in sports activities in order to gain more opportunities to socialize with their peers and have a space in which to discuss menopause issues openly. In addition, this study encourages menopausal women to get their own bodies moving, rather than just going to massage parlors and physiotherapy parlors to relieve discomfort using the services of others. Being physically active can also free more menopausal women from demanding housework and help them focus more on themselves, enriching their days and improving their quality of life. Additionally, it lets young women see the well-being of menopausal women, thereby reducing their fear and anxiety about aging.

### 5.3. Limitations

This study has certain limitations. First, the researchers did not account for moderator variables, such as health status and chronic diseases, in the study model. Future research should give more consideration to the potential for model changes and developments. Second, this study is cross-sectional, which limits its depth and breadth. Future research should use longitudinal research methods as much as possible and set up experimental control groups.

## 6. Conclusions

In response to the proposed research objectives, this study noted that approximately 45% of menopausal women experience different degrees of health anxiety. In addition, the results indicated that physical exercise, interpersonal competence, and emotional intelligence are important factors influencing health anxiety in menopausal women. In particular, physical exercise can affect health anxiety through the mediation of interpersonal competence and emotional intelligence. Therefore, this study recommends that the government, the community, and the family pay attention to the health problems of menopausal women and provide more opportunities for them to participate in physical exercise.

## Figures and Tables

**Figure 1 behavsci-13-00140-f001:**
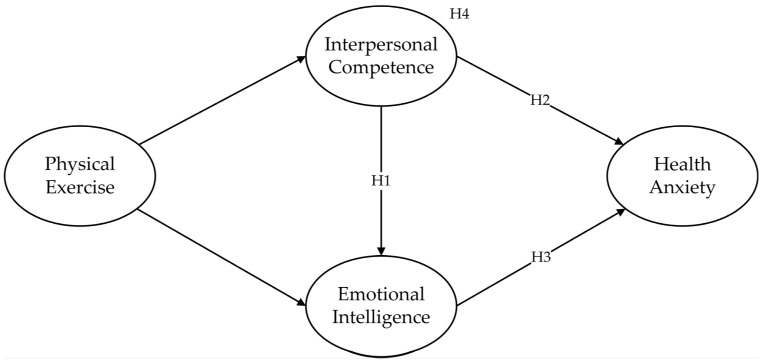
The hypothesized model.

**Figure 2 behavsci-13-00140-f002:**
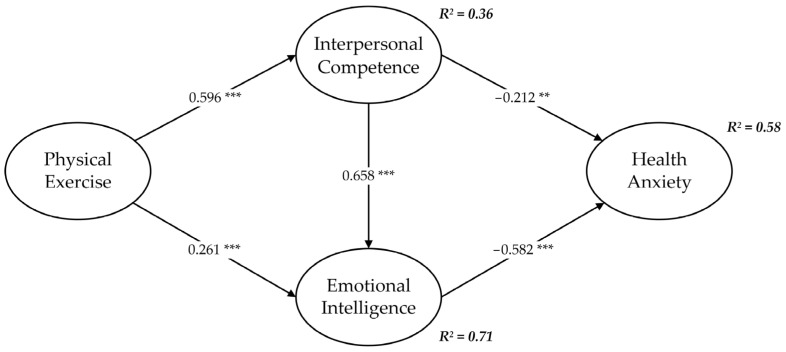
Structural path model. ** *p* < 0.01, *** *p* < 0.001. Standardized coefficients are reported.

**Table 1 behavsci-13-00140-t001:** Participant profile (N = 455).

Profiles	Survey (%)
*Age*	
40–45	32.1
46–50	54.7
51–55	13.2
*Menopausal status*	
Late reproductive stage	56.7
Menopausal transition	27.3
Postmenopausal	16.0
*Marital status*	
Married	87.3
Single (i.e., unmarried, divorced, or widowed)	12.7
*Education level*	
Below high school	33.4
High school/vocational school	61.1
College and above	5.5
*Employment status*	
Employed	63.1
Unemployed	36.9
*Income*	
<CNY 3000	29.2
CNY 3000–6000	49.5
>CNY 6000	21.3

**Table 2 behavsci-13-00140-t002:** Reliability and validity test.

Items	Loadings	Cα	AVE	CR
*Physical Exercise (PE)*		0.946	0.856	0.947
PE1	0.939			
PE2	0.922			
PE3	0.914			
*Interpersonal Competence (IC)*		0.929	0.769	0.930
IC1	0.815			
IC2	0.874			
IC3	0.897			
IC4	0.917			
*Emotional Intelligence (EI)*		0.917	0.787	0.917
EI1	0.891			
EI2	0.897			
EI3	0.873			
*Health Anxiety (HA)*		0.965	0.874	0.965
HA1	0.910			
HA2	0.931			
HA3	0.944			
HA4	0.953			

Note: Cα = Cronbach’s alpha; AVE = average variance extracted; CR = composite reliability.

**Table 3 behavsci-13-00140-t003:** Discriminant validity test.

Construct	PE	IC	EI	HA
PE	**(0.925)**			
IC	0.559 **	**(0.877)**		
EI	0.599 **	0.756 **	**(0.887)**	
HA	−0.576 **	−0.654 **	−0.704 **	**(0.935)**

Note: The square root of the average variance extracted (AVE) is in the diagonals (bold); off diagonals is a Pearson’s correlation of contracts. ** *p* < 0.01; PE = physical exercise; IC = interpersonal competence; EI = emotional intelligence; HA = health anxiety.

**Table 4 behavsci-13-00140-t004:** Standardized direct, indirect, and total effects.

	Point Estimate	Product of Coefficients		Bootstrapping				
				Percentile 95% CI		Bias-Corrected 95% CI		Two-Tailed Significance
		*SE*	*Z*	Lower	Upper	Lower	Upper	
*Direct effects*								
PE → IC	0.596	0.037	16.108	0.521	0.665	0.520	0.665	0.000 (***)
PE → EI	0.261	0.054	4.833	0.162	0.374	−0.160	0.371	0.000 (***)
IC → EI	0.658	0.051	12.902	0.549	0.749	0.549	0.748	0.000 (***)
IC → HA	−0.212	0.067	−3.164	−0.340	−0.074	−0.341	−0.075	0.004 (**)
EI → HA	−0.582	0.070	−8.314	−0.723	−0.447	−0.717	−0.441	0.000 (***)
*Indirect effects*								
PE → HA	−0.506	0.039	−12.974	−0.583	−0.428	−0.582	−0.427	0.000 (***)
*Total effects*								
PE → HA	−0.506	0.039	−12.974	−0.583	−0.428	−0.582	−0.427	0.000 (***)

Note: Standardized estimations of 5000 bootstrap samples. ** *p* < 0.01, *** *p* < 0.001; PE = physical exercise; IC = interpersonal competence; EI = emotional intelligence; HA = health anxiety.

## Data Availability

The data that support the findings and conclusions of this study will be available from the corresponding author upon reasonable request.

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
