# Peer review of "Keep Moving to Retain the Healthy Self: The Influence of Physical Exercise in Health Anxiety among Chinese Menopausal Women"

_behavsci, 2023, doi:10.3390/bs13020140_

Round 1
Reviewer 1 Report
Major concerns:
Abstract: The abstract states that the aim was to “investigate the interventional effect…,” yet, this is a cross-sectional study. Similarly, several times the words “affect” and “mediate” are used in the abstract. You are looking at relationships, not changes or effects of an intervention. These statements are misleading. Also, the use of surveys is not mentioned in the abstract.
Hypotheses: You’re not measuring impact (cause and effect). You are measuring relationships.
Minor comments:
Introduction, line 32. Did you mean “…women who suffer…?”
Introduction, line 62. “Different” is used two times in same sentence.
Introduction, line 63. People believe. Studies can’t believe.
Reviewer 2 Report
Please find attached the suggestions after reviewing this article

Reviewer 3 Report
This is a clear and well-written manuscript evaluating the impact of physical exercise on the health anxiety of menopausal women aged 40-55 in China. The content is original, the statistical analysis is broad and correct, and the number of validated questionnaires is large.
In general, the article is of interest to a wide audience, stressing the importance of providing the conditions for menopausal women to be involved in group physical activities to reduce health anxiety and chronic disease risk.
My suggestions concern some minor methodological aspects to be clarified/improved:
· Line 197: the reported percentage of respondents does not correspond to that in Table 1. Verify this.
· Lines 197-202: reporting the same numbers listed in the table is unnecessary. At most, indicate in which category the majority of respondents fall
· Line 236: add the acronym (CMV) after the expression "common method variation".
· Line 249: insert the appropriate expression "Average Variance Extracted" before the acronym to be reported in parentheses (AVE).
· Tables 3 and 4: report the meaning of the acronyms used in the first column in a note below the tables.
· Lines 346-351: Move the limitations of the study from the Conclusions to the last part of the Discussion, as is customary.
Round 2
Reviewer 1 Report
The authors did a commendable job of editing the manuscript to make it more clear and a better representation of the study.
Author Response
Thank you for your comment and affirmation. Have a nice day.
Reviewer 2 Report
I appreciate all the work done to address the suggestions, and simply hold two comments for your consideration. Congratulations on your work.
